# Treatment of Reticular Oral Lichen Planus with Photodynamic Therapy: A Case Series

**DOI:** 10.3390/jcm12030875

**Published:** 2023-01-22

**Authors:** Magdalena Ewa Sulewska, Jagoda Tomaszuk, Eugeniusz Sajewicz, Jan Pietruski, Anna Starzyńska, Małgorzata Pietruska

**Affiliations:** 1Department of Periodontal and Oral Mucosa Diseases, Medical University of Białystok, ul. Waszyngtona 13, 15-269 Białystok, Poland; 2Department of Biocybernetics and Biomedical Engineering, Białystok University of Technology, ul. Wiejska 45c, 15-351 Białystok, Poland; 3Dental Practice, ul. Waszyngtona 1/34, 15-269 Białystok, Poland; 4Department of Oral Surgery, Medical University of Gdańsk, Dębinki 7, 80-211 Gdańsk, Poland

**Keywords:** reticular oral lichen planus, photodynamic therapy, aminolevulinic acid

## Abstract

Objectives: The aim of the study was to clinically evaluate the efficacy of photodynamic therapy in treatment of the reticular form of oral lichen planus (OLP). Materials and Methods: Twenty patients aged 40–76, with 40 confirmed OLP lesions in total, underwent photodynamic therapy (PDT) following the authors’ own protocol, which used 5% 5-aminolevulinic acid as a photosensitizer applied two hours prior to illumination with a diode lamp emitting light at 630 nm and 300 mW. The therapy comprised of 10 weekly illumination sessions and was clinically evaluated between its completion and the end of a 12-month follow-up. Results: While the baseline mean size of all 40 lesions was 2.74 ± 3.03 cm^2^, it was 2.97 ± 3.4 cm^2^ for the 30 lesions on the buccal mucosa and 2.02 ± 1.32 cm^2^ for the remaining 10 on the gingiva and tongue. On completion of the therapy, 37 sites improved, including 14 showing complete remission. From that point, the mean size reduction of 56.2% (1.2 ± 1.4 cm^2^) rose to 67.88% (0.88 ± 1.3 cm^2^) 12 months later. Conclusions: The results suggest that ALA-mediated photodynamic therapy was effective for the reticular form of OLP and may become an optional or complementary treatment.

## 1. Introduction

Oral lichen planus is a chronic disease of the skin and mucosa of still scarcely known etiology [1], a fact that hinders effective treatment and makes it solely symptomatic, so any alleviated discomfort or relatively long-term remission is often regarded as a successful outcome.

The available literature does not offer clear guidelines for treatment of OLP [2,3,4,5,6]. The limited effectiveness of pharmacological treatment with routinely used corticosteroids, retinoids, and immunomodulators, burdened with side effects and frequent relapses, is pushing clinicians to seek alternative solutions [1,4,7,8,9,10]. Non-pharmacological therapies, such as surgery, cryosurgery, and laser treatment, are invasive, limited, or simply impossible due to a lesion’s type, extent, or location [5,11,12,13,14]. Topical PUVA (Psolaren Ultra-Violet A) and PUVASOL (Psolaren Ultra-Violet A and Solar Radiation) photochemotherapies or narrow-band UVB-NB-UVB cause multiple adverse effects and, most importantly, are potentially carcinogenic [15,16,17,18,19].

Recent years have seen attempts to use photodynamic therapy (PDT) to treat symptomatic OLP. It is only minimally invasive and can be used exclusively and repeatedly or combined with other therapeutic approaches. PDT involves the use of a photosensitive agent which first selectively adheres to the affected tissue and, when activated by light at a certain wavelength, uses its immunomodulating and cytotoxic power to destroy inflammatory cells [9,20,21,22]. The approach seems to be effective, but has so far been scarcely documented and implemented in varying protocols, which makes for discrepancies in studies.

This is why the aim of the present study was to assess the effectiveness of the authors’ protocol of ALA-mediated PDT in the treatment of the symptomatic reticular form of OLP. The secondary objective was the patients’ subjective evaluation of the treatment.

## 2. Materials and Methods

The therapy and its 12-month follow-up stage took place at the Department of Periodontal and Oral Mucosa Diseases, Medical University of Białystok, from 2016 to 2018. All patients gave their informed consent for inclusion before they participated in the study. The study was carried out in accordance with the Helsinki Declaration of 1975, as revised in 2013, and was reviewed and approved by the local ethical committee (Ethics Committee No.: R-I-002/263/2012).

A group of 20 patients that underwent PDT comprised 13 women and 7 men aged 40–76 (mean 58.38 ± 10.51) with an overall number of 40 OLP lesions among them. No patients were smokers at the time of the study, but 3 men admitted to smoking up to 20 cigarettes a day in the past.

The primary inclusion criterion for treatment was a histopathologically confirmed clinical diagnosis. Those who declared or confirmed any of the following facts or states were excluded from the study: age under 18, pregnancy, breast feeding, systemic diseases, allergy, drug use, photosensitivity, previous treatment for OLP at least 6 months prior to the beginning of the study, schedules conflicting with the regular weekly and monthly appointments, physical or mental disability that prevented keeping an occlusive dressing in place, or suffering from oral conditions other than OLP.

### 2.1. Therapeutic Procedure

The authors’ protocol incorporated 5-aminolevulinic acid (ALA) as a photosensitizer in a topical, semi-solid preparation with 5% concentration of ALA (Ala-Plus, Farmapol, Polska). An initial 2 mm layer of the preparation was applied directly onto the saliva-free lesion and surrounding mucosa 2 h before light activation. The site was protected with a non-woven occlusive dressing overlapping the lesion and stabilized with a few extra layers of sterile gauze. Three subsequent applications of the photosensitizer then followed every 30 min within 2 h. To activate the ALA, we used a custom-designed diode lamp providing light at 630 nm and 300 mW. The light had a peak power density of 120 J/cm^2^ at the spot area. The course of treatment comprised 10 weekly PDT sessions.

### 2.2. Examination

Prior to the initial appointment, the lesions were photographed and macroscopically measured in millimeters with the use of a periodontal probe PCPUNC 15 (Hu-Friedy, Chicago, IL, USA). Each lesion’s surface area was calculated by taking the distances between the farthest points bordering the healthy mucosa in the width and length of the lesion. For the purpose of this study, the lesions were grouped size-wise as follows:Size Group 0—no apparent lesion;Size Group I—lesions < 3 cm^2^;Size Group II—lesions in the range of 3 to 6 cm^2^;Size Group III—lesions > 6 cm^2^.

The patients’ subjective evaluation of the treatment was collected at every appointment using a Visual Analogue Scale [23]. Macroscopic examination and photographic documentation took place at every other appointment before ALA application. After the 10-week-long therapy proper, the patients entered a 12-month-long observation stage, with appointments scheduled every 3 months.

### 2.3. Statistical Analysis

Normal distribution of the parameters analyzed in the paper was verified by the Kolmogorov–Smirnov and Shapiro–Wilk tests with a Lilleforce correction. None of the parameters showed normal distribution. The Wilcoxon matched-pairs signed-ranks test was used to compare two related parameters. The means were calculated with a confidence interval of 95%.

The linear regression model was used to verify a relationship between age and each parameter. Additionally, a Pearson correlation coefficient was calculated. Levels of significance for differences between parameters in relation to gender and type of lesion were verified using a non-parametric U Mann–Whitney test, and the significance of differences for multiple comparison was calculated using a non-parametric Kruskal–Wallis test. Statistical significance was determined at *p* < 0.05. All calculations were conducted with Statistica 12.0 software (StatSoft, Tulsa, OK, USA).

## 3. Results

Out of all 40 lesions, 30 were within the lining mucosa, of which 28 were buccal and 2 were sublingual. The other 10 lesions were on the masticatory mucosa—8 gingival and 2 lingual.

The mean size of the lesions on the lining mucosa was greater than those in masticatory areas and, likewise, was greater in men than women (Table 1).

A series of 10 PDT sessions brought a statistically significant mean size reduction, the proportion of which was nearly fivefold greater for the lesions on the lining mucosa than for the lesions in masticatory regions. As further healing took place over the consecutive 12 months, the overall mean size reduction rose to 67.88% (Figure 1). The results attained at the end of the observation stage showed that PDT proved twice as effective in lining mucosa than in masticatory areas (Table 1 and Table 2).

On completion of the therapy, 37 lesions showed improvement, of which 14 on the lining mucosa fully healed. Three lesions—two gingival and onw buccal—did not respond to the therapy. There were five significantly smaller and painless recurrences on the buccal mucosa.

With regard to buccal and sublingual lesions, those in Size Group I showed the most remarkable mean size reduction directly after treatment, while those in Size Group II were reduced the least. At the end of the follow-up, only lesions originally in Size Group III indicated ongoing healing.

Among the lesions on the gingiva and tongue, those in Group Size I demonstrated the biggest mean size reduction, while those in Group Size III demonstrated the smallest reduction. At the end of the follow-up stage, ongoing healing was evident across all size groups, with lesions in Groups Size III and I reduced by nearly 40 and over 55%, respectively.

A year after the therapy, only three persistent lesions remained within Group Size III, four healed completely, two progressed to Group II, and three progressed to Group I, which now encompassed 17 lesions. The mean size reduction was greater in men than in women both on completion of the therapy and 12 months later. The difference was not statistically significant. Figure 1, Figure 2, Figure 3 and Figure 4 show four patients in whom OLP lesion treatment effectiveness varied from high to low.

The results show that PDT was more effective for buccal and sublingual lesions among men and reversely so in masticatory areas, a fact of no statistical significance. Linear regression analysis showed no relationship between the therapy outcome and the patient’s gender, age, or lesion location.

The Kruskall–Wallis Test proved that the lesions’ baseline size had a statistically significant impact on outcome—r = 0.74 at *p* < 0.0001 after 10 PDT sessions and r = 0.42 at *p* = 0.008 after 12 months.

Prior to PDT, the Visual Analogue Scale ranged from 3 to 8 with a mean of 6.0 ± 1.43. After the third session all patients reported intense pain lasting up to 2 h after treatment—the VAS ranged from 5–10, with a mean of 7.85 ± 1.42. After the 10th session, the VAS ranged from 0–6 with a mean of 3.4 ± 1.81, and a year later, it ranged from 0–5 with a mean of 2.5 ± 1.73.

## 4. Discussion

The aim of the study was to clinically assess the efficacy of our own protocol for treatment of the symptomatic reticular form of OLP. Directly after a series of 10 illuminations, 92.5% of the lesions had either healed or become smaller. The mean reduction was 56.2%, which was significant, reaching an even greater proportion of 65.66% on the buccal mucosa as compared to 13.86% on the gingiva and tongue.

The hitherto available literature presents PDT as a relatively effective modality of OLP treatment. However, discrepant results of studies, presumably reflecting a variety of protocols, make their interpretation and comparison difficult or unfeasible [9,24,25,26,27,28,29,30,31,32,33]. The protocols vary mainly by the choice of photosensitizer and its practical application, as well as the source and dosage of light. The most frequently used photosensitizers are ALA, methylene blue, and toluidine blue.

The protocol presented here closely follows the one in Sulewska et al. [24], who incorporated both the same 5% ALA photosensitizer and its technique of application, the only difference being a higher 150 J/cm^2^ dose of energy. Their results also resembled those presented here—improvement was apparent in 87.9% of the sites, including 37.09% of sites that healed completely, thus contributing to a higher proportion of 62.91% in mean size reduction. The same authors had previously followed a corresponding protocol in the treatment of erosive OLP forms, where, after a series of 10 illuminations, the mean size reduction was only 8.05%. However, a closer analysis showed that improvement took place in the majority of cases—16 out of 22 lesions healed completely, yielding a 48.99% mean size reduction. Among six lesions that did not respond to treatment, five enlarged, and one did not change. Notably, healing continued over the successive 12 months, resulting in a mean size reduction of 69.13% [25].

Elsewhere, Maloth et al. [26] used a 98% solution of 5-ALA applied 30 min prior to illumination with a diode light at 420 nm. Eight out of 10 lesions responded to treatment, none of them fully healed, and the overall size reduction was 36.49%. Yet another study using methyl 5-aminolevulinate (MAL) reported significant improvement but did not provide any relevant figures regarding changes in size [27].

PDT protocols incorporating other photosensitizers have led to comparable results. In one instance, improvement occurred in 61.53% of sites, reduced overall by 44.3% after a single PDT session mediated with methylene blue activated by laser light at 632 nm and a light exposure dose of 120 J/cm^2^ [9]. In another study by the same authors, 80% of lesions responded positively to treatment [28]. Other studies using methylene blue (MB) attained improvement in 50% of sites, with a 53.3% reduction [29,30]. On the other hand, chlorine-6-mediated PDT improved 81.25% of lesions, leading to a 55% mean size reduction [31].

There are particularly interesting, but largely heterogenous, studies that juxtapose PDT and the currently prevailing corticotherapy in terms of their efficacy [32,33,34,35,36]. Mostafa et al. [37] report a significant decrease in both objective and subjective signs and symptoms after PDT as compared with topical steroids, while other authors conclude that PDT is just as effective as the topical steroids [26,38,39,40,41,42]. Those voices, however, are contradicted by authors claiming that corticotherapy leads to a more pronounced reduction in size and fewer relapses [43]. It is understandable, then, that faced with such inconclusive results in the cited studies, a meta-analysis by Jajarm et al. [44] did not indisputably confirm the efficacy of PDT and rather points to an inadequate number of studies and range of diverse protocols and often too-short observation times.

All studies concerning mucosal lesions share certain flaws arising from both inaccurate measuring procedures and natural obstacles in the oral cavity—the mobility of the cheeks and tongue, curvatures, and a multitude of varying lesion outlines.

An additional and purely technical issue in PDT is that it is practically impossible to apply a uniform dose of the photosensitive agent around varied locations. Although an experienced clinician, always the same, took utmost care to take measurements consistently, as well as to control a semi-solid preparation dose with a periodontal probe, we are fully aware of such technical shortcomings, which may limit our study.

In the majority of published studies, the observation stage took between 3 and 9 months, and only in three cases between 1 and 4 years [9,28,29,30,31]. Our own previous studies showed that PDT not only sustains its immediate outcome for a considerable length of time, but also allows for further improvement throughout ongoing healing [24,25], a tendency which the current study also shows. Althoughfive of the buccal lesions reappeared within a year of PDT completion, they were significantly smaller and painless. Simultaneously, the lesions of all sizes on masticatory mucosa diminished, as did the large lesions on the lining mucosa. Despite the fact that 12 months after the therapy a group of 36 lesions that had healed on its completion grew just by 1 to 37, the percentage of their mean size rose by 11.68%, reaching 67.88%. At that point, only 3 out of 11 lesions initially larger than 3 cm^2^ remained unaltered.

Stimulation of long-term remission seems to be a considerable advantage of PDT over corticotherapy, which has multiple adverse side effects [1,8,9,34,35,43,44,45,46,47,48,49,50,51]. Important studies by Kvaall et al. [27] and Rakesh et al. [52] showed that the long-term PDT outcome can last up to 4 years for a majority of patients. The randomized clinical trials of Mirza et al. [53] remain in opposition to the above conclusions. The authors reported that PDT treatment of OLP was significantly better in reducing pain and clinical severity than corticosteroids, but relapses were more frequent in the PDT group.

A literature review conducted in 2021 by Azzouzi et al. [36] showed that PDT can be used to treat large and recurrent lesions with minimal impact, as the procedure is simple and is highly accepted by patients. The authors concluded that use of PDT has been gaining in importance due to its minimal toxicity to healthy tissues and the effective reduction in OLP changes after PDT. Of all the 12 studies included in the aforementioned literature review, only one study [43] showed that traditional corticosteroids are more effective than PDT.

Hesse et al. [32] identified seventeen clinical studies investigating PDT of OLP. In all studies except for one [43], PDT was superior or equal to conventional treatment. The majority of patients experienced a slight burning sensation during light activation, but no serious adverse events were reported. The authors noted that conventional corticosteroids carry a risk of common local and systemic side effects, and that patients may develop drug resistance. A meta-analysis by He et al. [39] concluded that PDT is as effective as topical corticosteroid in the treatment of OLP and could be used for cases resistant to steroids or when steroids are contraindicated. Similar conclusions were reported by Zborowski et al. [54]. In situations of topical or general contraindications to oral corticosteroids, resistance to them, or the need for repeated treatment in a short period of time, PDT appears to be a very promising treatment option. Since long-term use of systemic corticosteroids causes severe side effects, Saleh et al. [42] also recommended photodynamic therapy (PDT) as an alternative treatment to OLP. Separate studies by Cosgarea et al. [55] and Tocut et al. [33] identified PDT as a novel, safe, and non-invasive therapeutic option in OLP treatment. Additionally, Cosgarea et al. [55] reported that PDT treatment of OLP leads to lesion reduction, improvement in QOL (quality of life), and induces local and systemic anti-inflammatory effects.

In addition to PDT’s efficacy, our study explored the patients’ own perception of treatment as indicated by VAS. We were not only interested in assessing the intensity of pain in the context of healing lesions after treatment, but also the intensity of pain during PDT. The presented protocol included a 30 J lower energy dose compared to those we used earlier [24,25]. The purpose of reducing the dose was to limit unpleasant sensations during and immediately after PDT sessions, which were reported by previously treated patients. On the other hand, we wanted to maintain the effectiveness of the previously used protocol, in which a higher dose of energy was applied. Patients from the present study voiced their experience of discomfort during and after illumination sessions, but were not automatically given any anesthetic medication, and their perception of pain decreased with each illumination. After the 10th session, all patients suffered pain of lesser intensity, even though some lesions were still present. In the following 12 months, discomfort subsided further, but even when it was more palpable, it was substantially weaker than before therapy. A change in VAS values from 6.0 ± 1.43 to 2.5 ± 1.73 clearly shows that PDT has the potential for significant reduction in patients’ discomfort. Other authors write about either an instantaneous, PDT-induced pain lasting a few days [20,21,22,24,25,27] or patients not having experienced any therapy-related pain, burning, or other sensations during or after illumination [9,29,31]. According to most publications, however, patients who underwent PDT declared that they observed a considerable improvement in their well-being, as well as fewer and less painful relapses [9,24,25,27,28,29,30].

## 5. Conclusions

The results we attained prove that the protocol we adhered to ensures a successful therapeutic outcome, helps reduce patients’ discomfort, and improves the quality of their daily lives. Therefore, it can constitute an alternative or supplementary treatment for oral lichen planus.

## Figures and Tables

**Figure 1 jcm-12-00875-f001:**
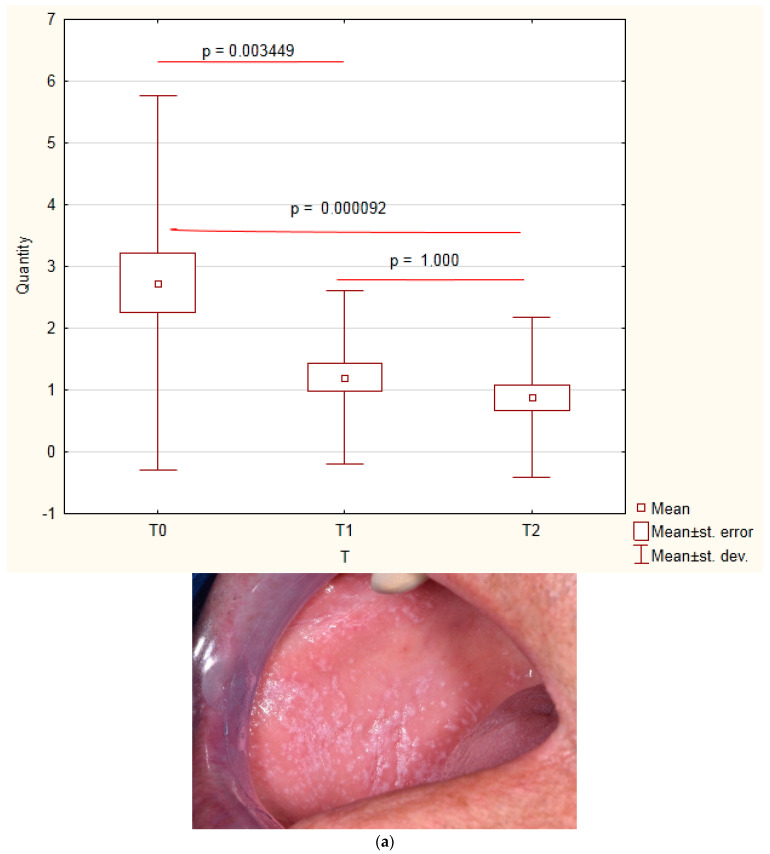
Changes in OLP over time. X: time of observation; T0: baseline; T1: after PDT; T2: 12 months after therapy. Y axsis: mean size of OLP lesions (cm^2^). (**a**) Male patient, aged 61, before PDT. OLP on the right cheek. Size of the lesion: 3 cm × 3 cm; surface area: 9 cm^2^. (**b**) After PDT complete remission of the lesion on the right cheek. (**c**) Twelve months post-op-remission on the right maintained.

**Figure 2 jcm-12-00875-f002:**
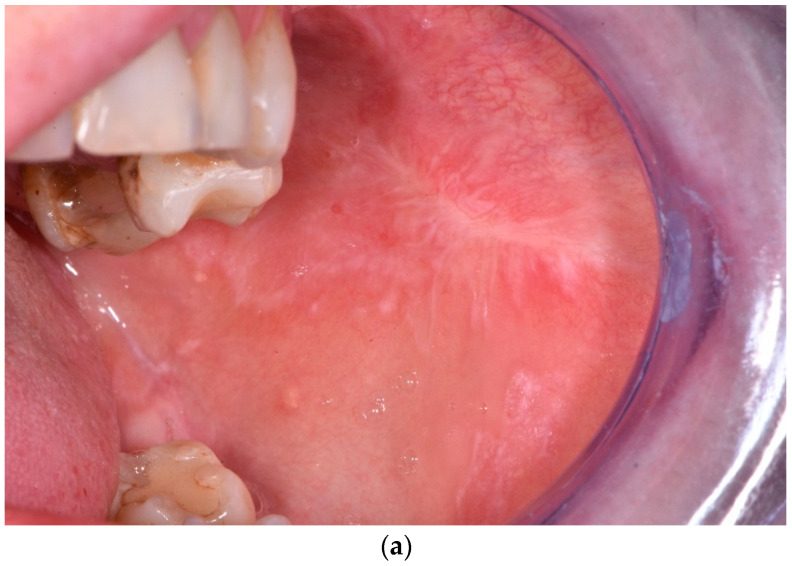
(**a**) Female patient, aged 44, before PDT. OLP on the left cheek. Size of the lesion—1.8 cm × 1.0 cm; surface area—1.8 cm^2^. (**b**) After PDT—lack of response to treatment. (**c**) One year later—extent of lesion area: 2.5 cm × 1.0 cm; surface area: 2.5 cm^2^.

**Figure 3 jcm-12-00875-f003:**
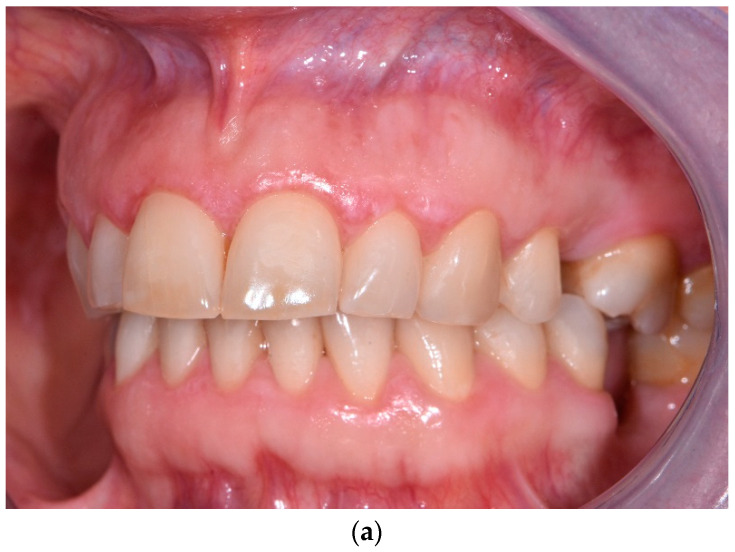
(**a**) Female patient, aged 44, before PDT. OLP on the right gingiva. Size of the lesion—1.1 cm × 0.7 cm; surface area 0.77 cm^2^. (**b**) After PDT—partial remission of the lesion. Size of the remaining lesion—1.1 cm × 0.5 cm; surface area—0.55 cm^2^. (**c**) Twelve months post-op—further improvement in OLP lesion. Size of the remaining lesion—0.5 cm × 0.6 cm; surface area—0.3 cm^2^.

**Figure 4 jcm-12-00875-f004:**
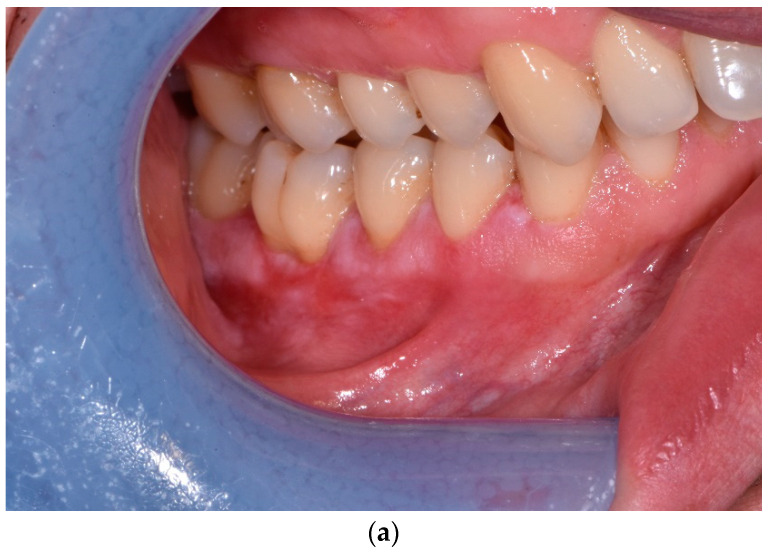
(**a**) Male patient, aged 40, before PDT. OLP on the left gingiva. Size of the lesion—2.5 cm × 0.6 cm; surface area—1.5 cm^2^. (**b**) After PDT—lack of response to treatment. (**c**) One year later—extent of lesion area: 2.5 cm × 0.8 cm; surface area: 2.0 cm^2^.

**Table 1 jcm-12-00875-t001:** Mean size, standard deviation, and mean reduction in OLP lesions at relevant stages of therapy.

Number of Patients	Location of Lesions	Number of Lesions	Mean Size (cm^2^) and Standard Deviation			Mean Size Reduction		*p*-Level of Significance	
			At Baseline	On Therapy Completion	12 Months after Therapy	On Therapy Completion	12 Months after Therapy	On Therapy Completion	12 Months after Therapy
20	Oral lichen planus	40	2.74 ± 3.03	1.20 ± 1.40	0.88 ± 1.30	56.20%	67.88%	0.003449 *	0.000092 *
13	Men	26	2.32 ± 2.61	1.15 ± 1.52	0.82 ± 1.29	50.43%	64.66%	0.000018 *	0.000027 *
7	Women	14	3.52 ± 3.66	1.30 ± 1.18	0.99 ± 1.35	63.07%	71.87%	0.003511 *	0.006947 *
15	Buccal and sublingual	30	2.97 ± 3.40	1.02 ± 1.40	0.75 ± 1.27	65.66%	74.75%	0.000006 *	0.000017 *
9	Women	18	2.35 ± 2.97	0.85 ± 1.48	0.61 ± 1.18	63.83%	74.04%	0.000196 *	0.000293 *
6	Men	12	3.79 ± 3.88	1.25 ± 1.24	0.95 ± 1.38	67.02%	74.93%	0.004742 *	0.013472 *
5	Gingival and lingual	10	2.02 ± 1.32	1.74 ± 1.32	1.27 ± 1.37	13.86%	37.13%	0.035693 *	0.011719 *
4	Women	8	2.22 ± 1.39	1.95 ± 1.44	1.40 ± 1.50	12.16%	36.94%	0.027709 *	0.027709 *
1	Men	2	1.86 ± 1.61	1.65 ± 1.06	1.26 ± 1.61	11.29%	32.26%	-	-

* statistical significance (*p* < 0.05).

**Table 2 jcm-12-00875-t002:** Mean size reduction, standard deviation, and 95% CI upon therapy completion and 12 months later.

Location of Lesions	Number of Lesions	Mean Size Reduction (cm^2^) at Baseline and on Completion (Mean ± Std.dev.)	Mean Diff. (95% Cl) between Baseline and Completion	Mean Size Reduction (cm^2^) at Baseline and 12 Months Post-Therapy (Mean ± Std.dev.)	Mean Diff. (95% Cl) between Baseline and 12 Months Post-Therapy
Oral lichen planus	40	1.53 ± 2.62	1.63 (0.72–2.53)	1.86 ± 2.83	1.96 (0.98–2.73)
Women	26	1.17 ± 2.37	2.37 (1.46–3.27)	1.50 ± 2.42	2.42 (1.49–3.35)
Men	14	0.97 ± 1.35	1.32 (0.64–1.99)	2.52 ± 3.48	3.39 (1.65–5.13)
Buccal and sublingual	30	1.95 ± 1.04	2.91 (1.87–3.95)	2.22 ± 3.16	3.17 (2.03–4.30)
Women	18	1.55 ± 2.77	2.72 (1.71–3.84)	1.81 ± 2.81	2.86 (1.83–3.89)
Men	12	0.95 ± 1.38	0.70 (0.23–1.17)	2.85 ± 3.67	2.35 (0.92–3.77)
Gingival and lingual	10	0.74 ± 0.91	0.74 (0.11–1.36)	0.86 ± 0.81	0.86 (0.29–1.42)
Women	8	0.60 ± 0.76	0.60 (0.07–1.13)	0.92 ± 0.91	0.92 (0.29–1.55)
Men	2	–	–	–	–

## Data Availability

Data available on request due to restrictions eg privacy or ethical.

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
