# Peer review of "Treatment of Reticular Oral Lichen Planus with Photodynamic Therapy: A Case Series"

_jcm, 2023, doi:10.3390/jcm12030875_

Round 1

Reviewer 1 Report

Dear Author,

Thank you very much for submitting your interesting work to JCM.

However, I am having several concerns regarding the design of the manuscript.

1.      Please explain why did you select reticular form of OLP, since it is usually asymptomatic? It would be better to select erosive or atrophic form of OLP, since it is painful and therefore desired therapy is required.

2.      I do not like the protocol for application of the ALA gel. I am not sure how come you obtained saliva free space 2h prior to laser application, and why did you wait for 2 hours.

3.      Why did you have three subsequent application of the photosensitiser?

4.      I think that 10 consecutive weeks of 3 hours per week (3 times application of photosensistizer, waiting for 2 hours, and aPDT application) is really unnecessary for asymptomatic OLP lesions. For this kind of OLP maybe application of corticosteroids is easier way of treatment.

Sincerely,

Reviewer

Author Response

Thank you very much for interest in our manuscript „ Treatment of reticular oral lichen planus with photodynamic therapy. A case series”.

We are very grateful for your time and honest evaluation of our work. 
All suggestions and comments were extremely valuable, so they were included in the revised version of the manuscript, significantly increasing its value.
Below we respond to Reviewer comments.

Sincerelly, 
dr. Magdalena Sulewska, 
prof. Małgorzata Pietruska

Thank you very much for submitting your interesting work to JCM.
However, I am having several concerns regarding the design of the manuscript.
1. Please explain why did you select reticular form of OLP, since it is usually asymptomatic? It would be better to select erosive or atrophic form of OLP, since it is painful and therefore desired therapy is required.

The reticular form of oral lichen planus is usually asymptomatic, but to our study included patients who reported pain and burning of the oral mucosa.
We have highlighted this fact in the section “Results” and “Discussion”.
In Results Section:
….Prior to PDT Visual Analogue Scale ranged from 3 to 8 with a mean 6.0±1.43. After the 3rd session all patients reported intense pain lasting up to 2 hours after treatment – VAS ranging 5-10 and mean 7.85±1.42. After the 10th session VAS ranged 0-6, mean 3.4±1.81 while a year later 0-5, mean 2.5±1.73…..
In Discussion Section:
…..After the 10th session all patients suffered pain of lesser intensity, even though some lesions were still present. In the following 12 months discomfort subsided further, but even when more palpable, it was substantially weaker than in pre-therapy time. A change in VAS values from 6.0±1.43 to 2.5±1.73 shows clearly that PDT has the potential for significant reduction of patients’ discomfort….
All patients felt discomfort due to the presence of lesions, and previously had the topical corticosteroids  treatment, however it did not bring expected results.
The time that had to elapse from steroid therapy was a minimum of 6 months because it was one of the conditions for inclusion in the study - as described in the "Materials and Method" section.
Looking for an alternative and non-invasive methods of treatment, we decided to use photodynamic therapy on those patients.
The reviewer's comment is correct, therefore the aim of our other research was to assess the effectiveness of photodynamic therapy in the treatment of erosive forms of oral lichen planus:
Sulewska M., Duraj E., Sobaniec S., Graczyk A., Milewski R., Wróblewska M., Pietruski J., Pietruska M. A clinical evaluation of the efficacy of photodynamic therapy in the treatment of erosive oral lichen planus: A case series. Photodiagnosis and Photodynamic Therapy 2017: 18, s. 12-19. doi: 10.1016/j.pdpdt.2017.01.178.

2.    I do not like the protocol for application of the ALA gel. I am not sure how come you obtained saliva free space 2h prior to laser application, and why did you wait for 2 hours.
3.    Why did you have three subsequent application of the photosensitiser?

Obtaining saliva free space in the mouth is a big challenge.
We tried to create the best possible conditions for the contact of the photosensitizer with the lesions of the OLP.
An initial layer of the topical semi-solid preparation with 5% concentration of ALA was directly applied to the lesion and the surrounding mucosa after salivation and wiping the surface with a sterile gauze pad. The site was protected with a non-woven occlusive dressing overlapping the lesion and stabilized with a few extra layers of sterile gauze.
The patient was seated in the dental chair with the saliva ejector turned on for saliva suction. 
The photosensitizer was applied 3 more times and the occlusive dressings were changed to ensure dryness thus prolonging the contact of the photosensitizer with the OLP changes in order to release it from the preparation.

Keeping in mind the specificity of the oral cavity environment - continuous production of saliva, decrease in concentration of the photosensitizer and thus reducing the possibility of releasing the photosensitizer, we decided to repeat the procedure 3 more times (2h prior to laser application) in order to obtain the best properties of the used ALA preparation.

However, we are aware that the duration of the procedure was long. Patients showed great patience and good cooperation.
Presented protocol was based on previous experience and satisfactory results of the therapy (Sulewska et al. 2017; Sulewska et al. 2019). 
The time-consuming nature of the procedure prompted us to look for a better solution.

We are currently working on novel type mucosa preparations for ALA delivery with improved bioavailability. Part of these studies is the assessment of the ability of the photosensitizer to penetrate into the deeper layers of the epithelium.

Sulewska M., Duraj E., Sobaniec S., Graczyk A., Milewski R., Wróblewska M., Pietruski J., Pietruska M. A clinical evaluation of the efficacy of photodynamic therapy in the treatment of erosive oral lichen planus: A case series. Photodiagnosis and Photodynamic Therapy 2017: 18, s. 12-19. doi: 10.1016/j.pdpdt.2017.01.178.
Sulewska M., Duraj E., Sobaniec S., Graczyk A., Milewski R.,Wróblewska M., Pietruski J., Pietruska M. A clinical evaluation of efficacy of photodynamic therapy in treatment of reticular oral lichen planus: a case series. Photodiagnosis and Photodynamic Therapy 2019: 25, s. 50-57. doi: 10.1016/j.pdpdt.2018.11.009

4.      I think that 10 consecutive weeks of 3 hours per week (3 times application of photosensistizer, waiting for 2 hours, and aPDT application) is really unnecessary for asymptomatic OLP lesions. For this kind of OLP maybe application of corticosteroids is easier way of treatment.

    It is true that the research was very time-consuming and required a lot of energy, time, discipline and cooperation of patients. Our patients were determined, very patient and motivated. 
Everyone reported improvement after the therapy, and the greatest satisfaction was the smile and contentment of our patients. 

All patients included in the study sought help with pain and had previously used corticosteroid therapy (at least 6 months earlier) which did not bring the expected results.
The corticosteroid therapy brought only short-term improvement.
Considering the chronic nature of OLP, achieving long-term remission should be considered a therapeutic success.
The obtained results brought satisfactory results not only immediately after the therapy, but they were maintained and were even better throughout the 12-month observation period.
On completion of the therapy 37 sites improved, including 14 showing complete remission. From that point, the mean size reduction of 56.2% (1.2cm2±1.4) rose to 67.88% (0.88cm2±1.3) 12 months later.
The results suggest that ALA-mediated photodynamic therapy was effective for reticular form of OLP and may become an optional or complementary treatment.

We are currently conducting a randomized clinical trial comparing the clinical effectiveness of photodynamic therapy  and topical corticosteroids in the treatment of oral lichen planus.
The results obtained after 6 months of observation are currently being prepared for publication and we hope that we will be able to present them to you in the near future.

Reviewer 2 Report

This is a well written and presented article. There are only a few points to consider. When there is more than one lesion in a patient would one expect similar reaction in all the lesions (intercolinarity).  

Table 1: Set significance level to 0.05 or 0.01 and mark with * those reading which are significant.

Table 2: there is no units in the head row of the table.

L 160 significant mean size reduction – should be:  The mean reduction was 56.2% and this was significant.   

L 224 Kvaall – misspelled: Kvaal

Fig 3b is upside-down

Fig 3c is upside -down

Author Response

Thank you very much for interest in our manuscript „ Treatment of reticular oral lichen planus with photodynamic therapy. A case series”.
We are very grateful for your time and honest evaluation of our work. 
All suggestions and comments were extremely valuable, so they were included in the revised version of the manuscript, significantly increasing its value.
Below we respond to Reviewer comments.

Sincerelly, 
dr. Magdalena Sulewska, 
prof. Małgorzata Pietruska

This is a well written and presented article. There are only a few points to consider. When there is more than one lesion in a patient would one expect similar reaction in all the lesions (intercolinarity).

Very often, when a patient has more than one lesion, we can expect a similar reaction of all lesions to the implemented treatment.
In our study, we observed intercollinearity in 19 out of 20 patients studied.
In a 74-year-old man, we observed a variable response to treatment for OLP lesions located on the buccal mucosa.
However, while one of the lesions responded to the treatment (after 10 PDT sessions, its size decreased from 2.25 cm2 to 0.2 cm2, and after 12 months of observation it was healed - remission), the second lesion - increased its size after the implemented therapy (after 10 sessions, her size increased from 2.25 cm2 to 3.04 cm2, reaching a size of 4 cm2 after one year of observation).

Table 1: Set significance level to 0.05 or 0.01 and mark with * those reading which are significant.
Suggested changes were included in the manuscript 

Table 2: there is no units in the head row of the table.
Suggested changes were included in the manuscript 

L 160 significant mean size reduction – should be:  The mean reduction was 56.2% and this was significant.   
Suggested changes were included in the manuscript 

L 224 Kvaall – misspelled: Kvaal
Suggested changes were included in the manuscript 

Fig 3b is upside-down
Suggested changes were included in the manuscript 

Fig 3c is upside -down 
Suggested changes were included in the manuscript